# Wastewater Purification with Nutrient and Carbon Recovery in a Mobile Resource Container

**DOI:** 10.3390/membranes11120975

**Published:** 2021-12-09

**Authors:** Hanna Kyllönen, Juha Heikkinen, Eliisa Järvelä, Lotta Sorsamäki, Virpi Siipola, Antti Grönroos

**Affiliations:** VTT Technical Research Centre of Finland, Ltd., P.O. Box 1000, 02044 VTT Espoo, Finland; hanna.kyllonen@vtt.fi (H.K.); juha.heikkinen@vtt.fi (J.H.); eliisa.jarvela@vtt.fi (E.J.); lotta.sorsamaki@vtt.fi (L.S.); virpi.siipola@vtt.fi (V.S.)

**Keywords:** wastewater, container, decentralized, filtration, precipitation, membrane, nutrient, recovery, hydrochar, techno-economic assessment

## Abstract

Water reuse from wastewater treatment plants can significantly reduce freshwater demand. Additionally municipal sewage and some industrial wastewaters could be used as sources of nutrients and carbon more effectively than they are used today. Biological treatments have attracted the most attention in wastewater purification, whereas, so far, only a little attention has been paid to the physico-chemical technologies. These technologies could, however, have great potential to recover nutrients when purifying wastewater. In this study, the main emphasis was to study the possibilities to utilize existing physico-chemical unit operations for wastewater purification and nutrients as well as carbon recovery. Unit operations were selected so that they could produce exploitable circular economy products from wastewaters and be assembled in a mobile container for carrying out recovery anywhere that is suitable. The results showed that in a mobile container, solids could be successfully separated from the studied wastewaters by flocculation-assisted solid/liquid separation and then processed into hydrochar by hydrothermal carbonization. Phosphate was precipitated using lime milk as calcium phosphate, and ammonium nitrogen was captured from the wastewater using membrane contactor technology resulting in ammonium sulphate for fertilizer use. Additionally, reverse osmosis retained residual impurities well, producing good quality water for reuse. The techno-economic feasibility seems promising.

## 1. Introduction

Recycled water is a valuable product of wastewater and can significantly reduce freshwater demand. It is especially important in water-scarce areas, such as in the city of Windhoek (Namibia), where 25% of the city’s potable water supply stems from wastewater, and in the city of Chennai (India), where the reuse of 40% of the generated wastewater satisfies 15% of the city’s water demand [1]. In Monterey (CA, USA), a large agricultural area is supplied with almost 80,000 m^3^ per day of nutrient-rich reclaimed municipal wastewater to irrigate and fertilize crops [1]. At the state level, Israel and Singapore are two examples of countries having nationwide wastewater reuse schemes. In Israel, almost a quarter of the country’s water demand is met by reclaimed wastewater, while Singapore achieves 40% with its NEWater reclamation plant [1]. Although a variety of activities have been conducted on water reuse, less have been conducted on advanced integration of nutrient and carbon recovery into water reuse. Municipal and also industrial wastewaters could be used more effectively as sources of nutrients and carbon than they are used today [2,3].

The ability to recover and reuse nutrients from wastewaters could reduce the dependency of energy-intensive processes used to synthesize mineral fertilizer products, such as nitrogen, and dependency on mined, finite mineral resources for phosphorous production. Nitrogen, a major component of fertilizers, is today manufactured using a Haber–Bosch process, which accounts for approximately 2% of worldwide energy demand [4]. Carbon recovery from wastewater would enable the production of energy products, soil improvers, and oil alternatives. The new EU Fertilizing Products Regulation will open the European market for recycled fertilizers, also for those originating from wastewaters and sewage sludges [5,6,7].

Usually, wastewater treatment is carried out in centralized wastewater treatment plants, but in sparsely populated areas, treatment of wastewater by decentralized treatment systems can decrease transportation costs of water and produce water for reuse and nutrients for recovery. Rapid population growth and industrialization has increased the necessity for decentralized treatment technologies. Decentralized solutions can also have other benefits compared to centralized systems, such as a lower footprint and easier maintenance [8]. They can be located near the sources of concentrated wastewaters. Thus, they could be a potential solution for wastewaters with valuables recovery. Septic tanks are the most common, and the least expensive, small-scale decentralized wastewater treatment systems, which require removal of sludge only once in 3 years, thereby generating very low operation and maintenance costs [8]. However, they are not sufficient for producing good quality water for reuse and much less for nutrient or carbon recovery.

In a conventional biological wastewater purification process, a large part of nitrogen is lost as gas and phosphorous as a part of awkwardly placed sewage sludge. Municipal sewage sludge contains phosphorous, but additionally pathogenic bacteria and viruses along with heavy metals, poorly biodegradable organic compounds, pharmaceuticals, and microplastics, which make its utilization management difficult [9]. However, several promising techniques for nutrient recovery from wastewater have been presented in earlier studies. These include, e.g., biological organisms, such as microalgae [10], or various chemical processes, such as ion exchange and sorption [11,12], struvite precipitation [13], calcium phosphate precipitation [14], electrochemical technologies [15,16], and membrane technologies [17,18,19,20]. Based on these studies, phosphorous precipitation at an elevated pH joined to ammonia gas recovery by membrane contactor technology were selected as the most potential technologies for nutrient recovery in a mobile container.

The use of calcium hydroxide for phosphate precipitation is enabled by the low cost and low toxicity of calcium hydroxide. Natural calcium orthophosphate (CaPO_4_) is the major source of phosphorus, which is used to produce agricultural fertilizers, detergents, and various phosphorus-containing chemicals [21]. Precipitation of phosphate as CaPO_4_ using a hydrated lime joined to particle separation could provide a simple, inexpensive method in a mobile container for retrieval from pre-treated wastewater with a low level of harmful impurities.

Recently, direct membrane filtration of wastewater in terms of different driving forces (such as pressure-driven, osmosis-driven, thermal-driven, and electrical-driven) has received much attention [19,20]. However, the application of direct membrane filtration in wastewater treatment and resource recovery is mostly limited by membrane fouling, especially in pressure-driven membrane processes. Therefore, novel membrane technologies have been applied for wastewater treatment. Hollow fiber liquid–liquid membrane contactor with gas-permeable membrane is an attractive technology for the recovery of volatile compounds, such as ammonia, having a low level of impurities. Ammonia gas can be separated efficiently from a solution at an elevated pH and temperature [5,22]. Thus, membrane contactor could be considered as a potential method for nitrogen separation as a part of an advanced mobile container recovery concept. Conventional membrane filtration, instead, could efficiently work as the last step in water production, when most of the impurities causing fouling have already been removed from the treated water.

The captured solids from wastewater can be used to produce bioenergy or hydrochar, which are both possible value-added products. Hydrothermal carbonization is a suitable technology to treat sludge containing a high amount of water that normally causes difficulties in transportation, as well as in conventional sludge treatment. Hydrothermal carbonization easily reduces mass and volume of waste sludges by producing a stable and sterile hydrochar product. Hydrochar has the potential to serve, for example, as fuel or soil amendment/fertilizer, or it can possibly be upgraded to higher value end uses such as, activated carbon [23]. A number of hydrothermal carbonization studies utilizing different sludges have been reported, but less have been reported for the sludge originating from physico-chemical unit operations.

In this study, a wastewater treatment concept was designed and constructed using physico-chemical unit operations, which enables separate water, nutrient, and carbon products with good reuse possibilities. The main emphasis was to determine and demonstrate the possibilities to utilize existing unit operations to design and construct a wastewater treatment system in a mobile container for the treatment of decentralized wastewater streams. When using a mobile resource container with physico-chemical technologies, wastewater treatment is easy to start up wherever found suitable, and the technologies can stand extreme conditions, such as low or high temperatures or pHs, and high nutrients concentrations. The technologies, which primarily can produce products having low impurity level, were in focus. Unit operations were confirmed at laboratory scale, and the best concept was scaled up and installed in a mobile container. The concept was piloted in two different applications with techno-economic assessment.

## 2. Materials and Methods

The studied physico-chemical unit operations to purify the wastewaters and recover the nutrients were: (1) chemically boosted solid–liquid separation by belt filtration; (2) microfiltration (MF); (3) membrane filtration; (4) precipitation; (5) membrane contactor (MC); and (6) hydrothermal carbonization (HTC). The performances of the separation processes were first studied at laboratory scale, using three different wastewater samples. Incoming wastewater from a municipal wastewater treatment plant (WWTP) was utilized as the primary feed water in concept building. Based on the laboratory tests, concept development was scaled up in a 20-foot mobile container and piloted using fish processing wastewater and septic tank wastewater from boats.

### 2.1. Characterization

Received wastewater samples and samples from each unit operations during the experimentation were characterized in terms of water quality and nutrient contents, i.e., nitrogen (N) and phosphorus (P); chemical oxygen demand (COD), which represented in this study organic carbon content in the wastewaters; pH; conductivity; total suspended solids (TSS); total solids (TS); and turbidity. A Hach DR3900 laboratory spectrophotometer (Hach, Ames, IA, USA) was used for the analysis of COD by cuvette methods LCK314, LCK114, or LCK014, depending on the COD concentration. The same spectrophotometer was also used to analyze total nitrogen (N total) (method LCK138), ammonium nitrogen (NH_4_-N) (methods LCK302 and LCK303), total phosphorus (P total), and orthophosphate (PO_4_-P) (method LCK350). Using method LCK350 without hydrolysis, only the (dissolved) PO_4_-P is measured, whereas with hydrolysis, all of the phosphorus is measured. Turbidity was measured using a HACH 2100AN IS Turbidimeter (Hach, Ames, IA, USA) and ISO method 7027. Conductivity was measured using a VWR Conductivity meter CO 3000 H (VWR, Germany). pH was measured using a VWR pH1000 pH-meter (VWR, Germany). The TSS content was analyzed by using standard procedure SFS-EN872 with Watman ME25 (0.45 µm) filter papers. The TS content was determined by drying an unfiltered sample in a glass beaker at 105 °C using standard procedure SFS3008. Elemental compositions of the hydrochars and activated carbons were determined using varioMAX CHN-analyzer (Elementar, Langenselbold, Germany) and Flash 2000-series elemental analyzer (Thermo Scientific, Waltham, MA, USA). Surface area and porosity analysis were performed using Micromeritics 3Flex analyzer (Norcross, GA, USA).

### 2.2. Chemical Treatment

The aim of flocculation and coagulation is to form larger-size clusters from particles in wastewater, facilitating faster and better solids removal by belt filtration. The flocculation and coagulation chemicals were tested alone and in combination with three different wastewater samples, using Kemira Flocculator 2000 equipment in 1 L beakers, with programmed sequencing of fast mixing (200 rpm for 30 s), slow mixing, i.e., floc formation (40 rpm for 5 min), and settling for 10 min before sampling. Tested flocculation chemicals from Kemira were powder Superfloc C-496 HMW and emulsions Superfloc C-1592RS, C-1594 and SD-6065, and from SNF Finland, powder FLOPAM FO4800SH. Active doses of the chemicals to wastewater varied from 1 to 20 ppm. Powders were assumed to be 100% of active substance, and a work solution of 0.1% concentration was produced for the Flocculator tests. The dry solids content was determined from emulsions, which assumed to be of active substance. Then, work solutions of 0.5% were prepared. Tested coagulation chemicals were Solenis’ Chargepac 9632 and Kemira’s Fennofloc F105. The main quality parameter to screen the flocculation and coagulation chemicals was water turbidity.

Phosphate removal using calcium hydroxide, i.e., lime (Ca(OH)_2_), was studied using municipal wastewater as a feed. Lime was added to wastewater as 10% lime milk. Tested doses varied from 100 to 1000 mg/L as Ca(OH)_2_, with a similar mixing procedure as with flocculants and coagulants. Phosphorous was determined from clarified solution. Since the phosphorous concentration in the piloted wastewater samples was low, lime was not used for phosphorous product recovery but only for increasing the pH in ammonia removal (see Membrane contactor).

### 2.3. Solid–Liquid Separation

Solid–liquid separation by belt filtration was studied in laboratory scale with a large number of available belts, using three different wastewater samples after flocculation. The tested belts’ pore sizes were 90, 131, 180, 250, and 350 µm. Belt filtering tests during the concept development and pilot scale studies were carried out using a Salsnes SF1000 belt filter unit, including an integrated dewatering system (screw) and a chemical feed system with 0.4–1 m^3^/h feed flows.

Solid–liquid separation by microfiltration was studied in laboratory scale with three different wastewater samples after belt filtration, using Watman FP3-9 3/4 filter housing with the following cartridge filters: (1) K01 PES 9 3/4” P0, 0.2 µm (absolute); (2) PURTREX PX01-9 3/4”, 1 µm (nominal); (3) PURTREX PX05-9 3/4”, 5 µm (nominal); and (4) PURTREX PX10-9 3/4”, 10 µm (nominal). In the pilot scale studies, SPE-5-9¾BB 5 µm (nominal) cartridge filters in a 20” Big Blue housing were used. The maximum feed flow was 1 m^3^/h.

### 2.4. Membrane Filtration

The performance of different membranes was evaluated by conducting laboratory scale filtration tests using a Millipore filtration device with three different wastewater samples. A Millipore filtration cell XFUF 076 01 (Millipore Corporation, Burlington, MA, USA), with agitation for increased shear on membrane surface, was used for laboratory scale membrane filtration tests with polymeric ultrafiltration, nanofiltration, and reverse osmosis membranes (UF, NF, and RO, respectively). The Millipore device consisted of a solvent resistant stirred cell designed for filtration applications having a cell volume of 300 mL. The maximum operating pressure was 6 bar, and the effective membrane area was 30 cm^2^. The Millipore system was equipped with data acquisition instrumentation and software, allowing real-time monitoring of pressure, temperature, and permeate flow during filtration experiments.

In the laboratory scale studies, the membrane filtration tests were conducted using flat sheet membranes from different manufacturers. The tested UF membranes UP150 and UP010 (Microdyn Nadir, Wiesbaden, Germany), which were made of polyethersulfone (PES), had molecular weight cut-offs of 150 kDa and 10 kDa, respectively. Tested NF membranes were NP030 (Microdyn Nadir) and NF270 (Dow Filmtec), and they had molecular weight cut-offs of 500–600 Da and 400 Da, respectively. Tested RO membranes were LG BW ES (LG Chem, Seoul, Korea) and LG SW (LG Chem). Pretreatment was carried out using a 4–12 µm filter paper in a Millipore test filter in laboratory scale tests. In the pilot-scale studies, membrane tests were carried out using a membrane unit having four LG BW 4040 ES RO membrane elements in four separate 4” pressure vessels BPV 4-300-EP 1 from Protec Arisawa. This membrane element has an effective membrane area of 7.9 m^2^; hence, the total membrane area was 31.6 m^2^. Pretreatment was carried out using a 5 µm cartridge filter. The feed flow in membrane filtration pilot studies was 0.4–1.0 m^3^/h.

### 2.5. Membrane Contactor

Ammonium (NH_4_^+^) recovery was studied using a membrane contactor (MC) and septic wastewater as a feed. In the laboratory scale studies, a 2 L septic wastewater was used as such, while in the pilot studies, a 90 L RO concentrate was used. Sulfuric acid worked as the receiving solution for the recovery of N by means of a reaction converting ammonia to ammonium sulfate ((NH_4_)_2_SO_4_). Before feeding the sample to the MC’s shell side, the pH was adjusted to >12 with lime slurry for transforming ammonia to NH_3_ gas form. In the laboratory scale studies, a commercial hydrophobic 0.2 µm flat sheet membrane MD QL816 was used, while in the pilot studies a commercial hollow fiber module 3M™ Liqui-Cel™ MM−1 × 5.5 Series X50 fiber was used.

### 2.6. Hydrothermal Carbonization and Activation Treatments

HTC technology is used to carbonize organic raw materials in water at moderate temperatures (~180 to 250 °C) under self-generated pressures (<50 bar) for a few hours. The aim of using HTC treatment in the pilot scale studies was to concentrate as much carbon as possible from the wastewater into a solid hydrochar product, to be further utilized, for example, as a raw material for activated carbon (AC).

The HTC experiments were performed using a high-pressure reactor (10 L) equipped with an electrical heater band, temperature control, digital and analog pressure indicator, internal water-cooling coil, and PC-controlled data logger. Solids were collected for the HTC experiments from both piloted concepts. The feeds were prepared before feeding to the reactor by mixing a desired amount of sludge and water, and adding 98 wt-% H_2_SO_4_ solution to adjust the pH of the sludge to a desired value. For the solids originating from the fish wastewater, two pH-values (3.5 and 2.2) and two treatment temperatures (240 °C and 270 °C) were used. For the solids originating from the septic wastewater, one pH-value (5.5) and one treatment temperature (220 °C) was used. The differences in the treatment conditions between the solids were caused by the sample type. The solids originating from the fish wastewater were estimated to require harsher conditions compared to the solids originating from the septic wastewater. The reactor was sealed and flushed with pressurized nitrogen, which was vented before the heat treatment. The samples were stirred during the heat treatment using 300 rpm speed. The reaction sequence consisted of reactor heating ramp to the set temperature, hold up of 180 min, and water cooling to ambient temperature before venting the residual pressure and opening the reactor.

The hydrochar produced from the septic wastewater solids was selected for the AC experiments due to the higher hydrochar yield. The hydrochar was chemically treated using potassium hydroxide (KOH) or phosphoric acid (H_3_PO_4_), and subjected to heat treatment under inert atmosphere (N_2_). Two different chemical:hydrochar ratios were used with KOH (1:1 and 2:1), and one with H_3_PO_4_ (1:1). The KOH impregnated hydrochars were heat-treated in 800 °C for 1 h. For the H_3_PO_4_ impregnated sample, heat treatment conditions of 450 °C and 1 h were used. After the heat treatment, the produced activated carbons were washed with diluted hydrochloric acid and ionized water to remove excess chemical from the pores. The washed and dried activated carbons were analyzed for their surface area and pore size distributions.

### 2.7. Mobile Container

Based on the laboratory scale studies, a mobile container was built up and tested with fish processing and septic tank wastewaters. A mobile container enables decentralized wastewater treatment which decreases transportation costs of wastewater. On the other hand, physico-chemical technologies in a resource container simplify startup of wastewater treatment and make extreme conditions, such as low or high temperatures or pHs, and high nutrients concentrations, possible. The pilot system was constructed in a 20-foot container having 1 m^3^/h treatment capacity. It consisted of the following separation equipment: (1) Salsnes SF1000 belt filter with an integrated screw and a polymer feed system; (2) two parallel SPE-5-9¾BB 5 µm (nominal) cartridge filters in a 20” Big Blue housing, and (3) a RO membrane unit including four LG BW 4040 ES (LG Chem) membrane elements in four separate 4” pressure vessels (BPV 4-300-EP 1, Protec Arisawa), and chemical addition and cleaning-in-place (CIP) options (Figure 1).

The container had online measurement devices for temperature (RTD Thermometer TR24, −50 to +400 °C), pressure (Cerabar PMP21, 25 bar), flow meters (3 units of Prosonic ultrasonic clamp-on flow sensors), and conductivity (Watman’s RO permeate conductivity meter). Ecograph T RSG35 with eight input signals was used in data manager and as a display unit. Therefore, real-time data display consisted of flow measurements of belt filter feed, RO feed and RO permeate, RO pressure, wastewater temperature, and RO permeate conductivity.

### 2.8. Techno-Economic Assessment

For evaluating the treatment (EUR/m^3^ wastewater) and investment costs (EUR) for both piloted concepts, a conceptual-level techno-economic assessment (TEA) was conducted. The overall production concept was designed for the treatment of the fish processing wastewater and septic tank wastewater. The process parameters (i.e., reduction figures and chemical consumption figures) in each separation step were gathered from the pilot scale tests. The mass and electric energy balances were calculated using spreadsheet balancing (MS Excel).

The block diagrams for the fish wastewater and septic wastewater concepts are shown in Figure 2 and Figure 3. Two cases were evaluated for both concepts. The process design was based on the mobile container pilot trials. In the case of FishWW#1, the target was to cut down the COD level of the wastewater, and thus earn some savings in the wastewater treatment fees. The wastewater is filtered with a belt filter (131 µm) to separate most of the TSS and to cut down the COD level. The separation is boosted with a flocculant (7 L/m^3^) addition. The filtered solids are further pressed with an integrated screw to produce a solid residue (average TS 67.5%), which may be treated in biogas production. The COD-reduced wastewater is directed to a municipal WWTP. In the case of FishWW#2, the target was to produce pure water for reuse or discharge. The COD-reduced water from the FishWW#1 case is further treated in a cartridge filter unit (5 µm) to remove the rest of TSS. There are two parallel cartridge units operating, while two are under replacement. Biocide (16 ppm) is added to protect the membranes in the RO unit, where nearly all dissolved solids are removed. The RO unit used in TEA consists of four BW 8080 ES membrane elements. The RO concentrate is directed to a municipal WWTP. The purified water may be reused, for example, as washing water in the fish processing plant. Because the concentrations of NH_4_-N and PO_4_-P in the feed wastewater are low, they are not recovered as separate streams. The production of biogas from the suspended solids or the fee to be paid for discharging the RO concentrate to the municipal WWTP, are not included in the TEA.

In the SepticWW#1 case, the wastewater is filtered with a belt filter (350 µm) to separate most of the TSS. The separation is boosted with a flocculant (90 L/m^3^) addition. The filtered solids are further pressed with an integrated screw to produce a solid residue (average of TS 20%), which may be treated in HTC or biogas production. The filtered water is further treated in a cartridge filter unit (5 µm) to remove the rest of the TSS. There are two parallel cartridge units operating, while two are under replacement. Biocide (16 ppm) is added to protect the membranes in the RO unit, where mainly all dissolved solids are removed. The RO unit used in TEA consists of four BW 8080 ES membrane elements. The RO concentrate is directed to a municipal WWTP. The purified water may be discharged back to sea. Because the concentration of PO_4_-P in the feed wastewater is low, it is not recovered as a separate stream. The recovery of NH_4_-N from the RO concentrate by membrane contactor was studied in pilot scale, but it is not included in the TEA. In the SepticWW#2 case, compared to SepticWW#1 case, the volume flow through the treatment system is decreased, and there are two smaller RO units in series. Both RO units consist of four BW 4040 ES membrane elements. The RO permeate from the first RO unit enters the second RO unit. The combined RO concentrates are directed to a municipal WWTP. Due to the smaller volume flow, there is also only one cartridge unit operating, while another is under replacement. The purified water may be discharged back to sea. The production of biogas or hydrochar from the suspended solids or fee to be paid for discharging the RO concentrate to a municipal WWTP are not included in the TEA.

The fish production plant, studied here, produces approximately 60 m^3^ of wastewater daily and operates 250 days annually, totaling an annual wastewater amount of 15,000 m^3^. In the FishWW#1 case, the wastewater treatment system is operated batch-wise two times a day, one hour at a time, five days a week (total 500 h/a, uptime 5.7%). The wastewater flow to the treatment system is 30 m^3^/h. In the FishWW#2 case, the wastewater treatment system operates continuously 104 h a week (5200 h/a, uptime 59%). The wastewater flow to the treatment system is 2.9 m^3^/h. In the FishWW#1 case, the wastewater treatment unit is placed inside the fish processing plant whereas in the FishWW#2 case, it is placed inside a mobile 20-foot container. The existing production staff at the fish processing plant are assumed to operate the treatment system; there are no additional labor costs.

The septic wastewater is collected from boats and delivered to the wastewater treatment system once a week during the sailing season, which is assumed to last in Finland 28 weeks annually. The volume of each delivery is approximately 20 m^3^. The annual septic wastewater amount is thus 560 m^3^. In the SepticWW#1 case, the treatment system operates continuously six hours once a week during the sailing season (168 h/a, uptime 1.9%). The wastewater feed to the treatment system is 3.3 m^3^/h. In the SepticWW#2 case, the treatment system operates 20 h during the 28 weekends (560 h/a, uptime 6.4%). In this case, the wastewater feed to the treatment system is 1 m^3^/h. The idea behind the smaller volume flow is the savings in the investment costs of the RO units. The wastewater treatment system is placed in both cases inside a mobile 20-foot container. In the SepticWW#1 case, the labor required, including both the operation and cleaning of the system, is eight hours, 28 times a year (224 h/a). For SepticWW2, the purification system is assumed to operate automatically for 20 h 28 times a year. Labor is needed each time only for four hours to start-up, shut-down, and clean the system (112 h/a).

The technical performance of the unit operations in the evaluated concepts were based on results from the pilot scale experiments. The equipment manufacturers were contacted to inquire about data on the capacity range, electricity consumption figures, equipment sizes, and purchased equipment costs. The following specific electricity consumption figures obtained from the literature and equipment manufacturers, and were used as: 0.7–1.2 kWh/m^3^ for the belt filter, including an integrated dewatering system (screw); 0.05–0.15 kWh/m^3^ for the cartridge filters covering the wastewater pumping; and 1–5.6 kWh/m^3^ for the RO unit. The filter inside the cartridge filter housing is changed and discharged once a week, meaning 50 and 28 annual replacements for the FishWW cases and SepticWW cases, respectively. The RO units consist of four 4” (7.9 m^2^/each) or 8” (40 m^2^/each) brackish water (BW) membrane elements, depending on the capacity of the treatment system. For capacities less than 1 m^3^/h, four BW 4040 ES membranes, with a total area of 31.6 m^2^, were selected. For capacities between 1 and 4 m^3^/h, four BW 8080 ES membranes, with a total area of 160 m^2^, were selected. The membranes require regular cleaning and occasional replacement. The membrane lifetime was assumed to be five years. They will be changed twice during the 15 years’ lifetime of the wastewater treatment container.

For evaluating the economic viability, variable costs and fixed costs were defined. Prices for calculating the variable costs, material costs deriving from the replacement of the cartridge filters and RO membranes, and the cleaning costs of the RO units are shown in Table 1. The fixed costs included labor costs (20 EUR/h), labor overheads (30% of labor costs), maintenance (3% of the fixed capital investment), and taxes and insurances (1% of the fixed capital investment).

Equipment cost estimates from equipment suppliers and literature were used to obtain the total purchased equipment cost (PEC) estimate for both concepts. The fixed capital investment (FCI) was then estimated using the detailed factorial method, where the PEC is multiplied by the appropriate factors [24]. The factors are listed in Table 2. They are based with minor modifications on the factors given for a solid–liquid process built from carbon steel [23]. Because the wastewater treatment system is placed either inside an existing plant or inside a mobile container, the factors for structures and buildings (f_6_) and offsites (f_9_) are ignored. Taking into account the material factor (f_8_) from carbon steel to stainless steel, the detailed factorial method suggests that the FCI is 3.61 times the PEC. This coefficient is called the installation factor. Lang proposed, based on 1940s economics, a value of 3.63 for mixed fluid–solids processing plants [24], which is close to the estimate above. However, discussions with equipment suppliers who provide turnkey solutions for mobile pure water production systems placed inside containers revealed that a coefficient of 1.3 to 1.8 is more realistic, particularly for container scale applications. As a result, an installation coefficient of 1.55 was used in this study.

Finally, the total capital investment (TCI), including the FCI and the working capital investment (WCI) was calculated. In this study, the WCI was ignored due to the simplicity of the process. This led to the TCI equaling the FCI. The FCI is presented as such without converting it into an annual capital charge (i.e., annualized capital cost) and without including it to total treatment costs. A sensitivity analysis was performed to reveal the influence of the main parameters affecting the total treatment costs in the SepticWW#2 case by varying key cost parameters: share of maintenance, taxes, and insurances of the FCI (−75% to +75% of base case value), and capacity (+25 to +1000% of base case value).

## 3. Results

### 3.1. Preliminary Studies at Laboratory Scale

#### 3.1.1. Characterization

In water characterization, the main studied parameters were the amount of TSS, COD, N total and NH_4_-N, and P total and PO_4_-P. Table 3 presents the water characterization results of the studied wastewater samples. The municipal wastewater was utilized as a primary feed water in the concept building. The fish processing wastewater was more suitable for carbon product and pure water production due to the low NH_4_-N and PO_4_-P concentrations and high COD concentration. Apart from water reuse and carbon recovery, septic wastewater was also suitable for nitrogen recovery.

#### 3.1.2. Chemical Treatment and Solid–Liquid Separation

The municipal wastewater was used as a starting point for the concept development at laboratory scale. Flocculation combined with belt filtration removed successfully suspended solids (SS) and COD from the municipal wastewater (Figure 4).

At laboratory scale, the studied flocculation chemicals for the fish processing wastewater were cationic polyacrylamides K5060, K2120T, K7500K, and anionic polyacrylamide A305 from Kemira, and cationic polyacrylamide FLOPAM FO4800SH from SNF. The rejection of turbidity of the best flocculants with three doses are shown in Figure 5.

At laboratory scale, the tested flocculation chemicals for the septic wastewater were Superfloc C-496 HMW from Kemira and FLOPAM FO4800SH from SNF. Both tested flocculants performed adequately in TSS flocculation. When a wider FO4800SH dose range was tested, an optimum dose of around 90 L/m^3^ (8 kg/t TS) of 0.1% solution, i.e., 90 ppm, was obtained. Since the septic wastewater contains a low concentration of nutrient PO_4_-P, coagulation was tested in order to find out if phosphorus could be precipitated into a solids fraction prior to belt filtration. Figure 6 shows the concentrations of COD, P total, NH_4_-N, as well as turbidity when the septic wastewater was chemically treated either using flocculation or coagulation–flocculation prior to belt filtration (350 µm). The tested coagulant was Solenis’ ferric chloride Chargepac 9632, and the dose was 1.5 L/m^3^, i.e., 1500 ppm as the product. By adding coagulant, the flocculant FO4800SH dose was reduced from 90 to 60 ppm (60 L/m^3^ of 0.1% solution; 5 kg/t TS).

Phosphorous recovery was not in the focus of the study due to the low P concentration of the studied wastewaters. However, precipitation of phosphate by lime was studied as an example of phosphorous recovery from wastewater. Lime removed the PO_4_-P from the municipal wastewater without any influence on the NH_4_-N concentration (Figure 7). Optimal phosphate removal occurred when lime addition increased the pH from 7.5 to above 10, suggesting that pH monitoring during the phosphate removal process could provide a simple method for ensuring consistent operation. However, the portion of PO_4_-P to added lime was quite low. For example, a dose of 500 mg Ca(OH)_2_ removed 10 mg of PO_4_-P. Therefore, the recirculation of lime milk should be investigated.

#### 3.1.3. Membrane Filtration

During laboratory scale membrane filtration experiments for the municipal wastewater, UF membranes (MWCO 150 and 10 kDa) rejected about 15% to 18% of dissolved organic carbon, whereas the NF membrane (MWCO 400 Da) rejected 64% of dissolved organic carbon, 20% of dissolved nitrogen, and 100% of dissolved phosphorus. Therefore, NF could be used to separate the main portions of dissolved nitrogen and phosphorus from each other.

Laboratory scale membrane filtration tests for the fish processing and septic tank wastewaters for nutrient rejection and pure water production included NF and RO membranes with different molecular weight cut-off (MWCO). The tested membranes were: (1) NF270 (400 Da) from Dow Filmtec; (2) NP030 (500–600 Da) from Microdyn Nadir; and (3) LG BW ES (100 Da) and LG SW (100 Da) from LG Chem. RO membranes were sufficient for adequate COD, NH_4_-N, and PO_4_-P rejection from the fish processing wastewater (Figure 8a). Brackish water RO membrane’s (LG BW ES) rejections for NH_4_-N and PO_4_-P were 98% for the fish processing wastewater. Since NF membranes were not satisfactory for nutrient rejection for the septic wastewater (Figure 8b), both piloting concepts were planned to be conducted with RO membranes.

#### 3.1.4. Membrane Contactor

Before feeding the RO concentrate to the MC, the pH was adjusted to >11 with NaOH or 10% lime slurry for transforming ammonia to NH_3_ gas form. High pH was applied due to low temperature, 20 °C, during the MC experiments. At the same time, dissolved PO_4_-P was precipitated and filtered using a 0.2 µm cartridge filter. The NH_4_-N content of the MC feed was 1150 mg/L. The H_2_SO_4_ solution worked as the receiving solution. During the recovery, samples were collected, and NH_4_-N was analyzed from both solutions. Additionally, the pH and mass of acid and conductivity of the feed were also followed. In 27 h, the NH_4_-N content decreased 99% and a total of 85% of NH_4_-N was adsorbed in 9.8% H_2_SO_4_. Then, it was possible to produce a 3.2 g/L NH_4_-N solution, i.e., 15 g/L of (NH_4_)_2_SO_4_.

### 3.2. Piloting in a Mobile Container

#### 3.2.1. Fish Processing Wastewater

The pilot concept for the fish processing wastewater consisted of flocculation, belt filtration, cartridge filtration, and RO (see Figure 2).

In the pilot trials, excellent purification results were obtained. Using a belt filtering with an integrated screw, the separation, thickening, and dewatering of solids from flocculated wastewater could be performed in one step. Active doses of the flocculants varied from 1 to 20 ppm in the tests. FO4800SH was the best among tested chemicals, with optimum dose of 6 ppm. When the fish wastewater flocculated with FO4800SH at dose 6 ppm (equals 2 kg/t TS) was poured onto the belt (pore size 131 µm) in the pilot scale test unit, effective solid and water fraction separation was obtained. The TSS and turbidity reductions were over 99%, and the COD reduction was already up to 90% after the flocculated wastewater belt filtration being the main recovery method for carbon. The belt filter (131 µm) performed effectively when flocculation worked well. However, this was highly dependent on the feed wastewater quality, which varied quite a lot during the piloting. The solid fraction obtained from the screw press was quite dry; the dry solids content was between 50% and 85%. RO pressure ranged from 3 to 5 bar, resulting in a flux of approximately 7 LMH. The water recovery (WR) ranged from 50% to 60%. The RO permeate contained 1.1 mg/L N total, 0.1 mg/L P total, and 15 mg/L COD being good enough either for reuse or discharge (Figure 9).

#### 3.2.2. Septic Wastewater

The pilot concept for the septic wastewater consisted of flocculation, belt filtration, cartridge filtration and RO (see Figure 3). The nutrient (NH_4_-N) recovery from the RO concentrate with MC was conducted separately.

The TS of the suspended solids fraction retained on the belt filter was 8% after the belt (350 µm) and 20% after the integrated screw press. Applied RO pressure ranged from 13 to 16 bar, resulting in a flux of approximately 5 LMH. The WR ranged from 30% to 50%. Figure 10 shows the concentrations of COD and nutrients (N total, NH_4_-N, P total, and PO_4_-P) in the incoming septic wastewater and after each purification step, as well as the rejection figures. The RO step concentrated the nutrients well and produced a permeate with fairly low concentrations of COD and P total, 76 mg/L and 0.6 mg/L, respectively. Instead, the RO permeate after the first RO step contained 79 mg/L N total and 77 mg/L NH_4_-N due to the high N concentration of the feed. It was too high and an additional RO step was carried out, i.e., two-pass RO, decreasing the N concentrations to 20 mg/L.

Before feeding the RO concentrate to the MC’s shell side, the pH was adjusted to >12 with 20% lime slurry for transforming NH_4_-N to NH_3_ gas form. High pH was applied due to low temperature, 14–18 °C, during the MC experiments. At the same time, dissolved PO_4_-P was precipitated. After 2 h settling, the overflow was filtered using a 1 µm cartridge filter and fed to the MC. PO_4_-P removal from RO concentrate was 99% and TSS content was half of the original after microfiltration with a 1 µm cartridge filter. The NH_4_-N content of the MC feed was 1760 mg/L. The feed solution volume was 90 L, and the amount of 0.5 M H_2_SO_4_ as the receiving solution was 9 kg. The concentration of the receiving solution was low, because the supplier of the hollow fiber membrane could not guarantee that the housing material (polycarbonate) could tolerate strong acid. The acid-receiving solution was fed to the MC’s lumen side. The flow rate of both feed and receiving solutions was 0.5 L/min (30 L/h), and the liquids were circulated for 225 h at a temperature of 14–18 °C. During the recovery, samples were collected, and NH_4_-N was analyzed from both solutions. Additionally, the pH and mass of acid and conductivity of the feed were also followed. During MC filtration, the conductivity of the feed decreased and the pH of the acid increased (Figure 11). NH_4_-N recovery from the feed decreased its ion content; thus, the conductivity decreased. Meanwhile, ammonia neutralized acid, which increased the pH of the receiving solution. Because of the pH increase in the receiving solution, acid was added to improve the NH_4_-N recovery from the feed to receiving solution. When piloting the septic wastewater, 82% of ammonia was absorbed in the receiving solution from the RO concentrate using MC. It was possible to produce a 12 g/L NH_4_-N solution, i.e., 57 g/L of (NH_4_)_2_SO_4_.

#### 3.2.3. Hydrothermal Carbonization and Activation Treatments

The solids fraction obtained by belt filtering the fish wastewater foamed extensively during the HTC reaction. The foaming was probably caused by the high protein and lipid content of the feedstock. The foam floated on top of the water phase, and after drying, consisted mostly of oil. The hydrochar yield was very low. The modification of the pH and temperature only slightly reduced the amount of oil. The major product of the HTC reaction with both test conditions was therefore oil, the so-called biocrude liquor. The produced hydrochar was not tested for activated carbon production due to the low yield.

The HTC treatment of the solids fraction obtained by belt filtering the septic wastewater produced a brownish lignite-like hydrochar. The hydrochar yield was very high, 89.8%. The carbon content of the hydrochar was 62 wt-%, which is quite typical for hydrochar, but also reflects the high ash content of the feed material: the ash content of the hydrochar was 13 wt-%. The amount of oxygen in the hydrochar was also fairly high (12.6 wt-%, calculated as 100%-(C + H + N + S + ash) wt-%). Higher oxygen contents may indicate a higher surface activity on the hydrochar, which makes it an interesting precursor for activated carbon. High oxygen content can increase its reactivity to activation chemicals and produce highly surface-active activated carbons. The carbon content of the hydrochar reflects the carbon content of the feed materials but is also affected by the treatment conditions.

The activation tests using the hydrochar originating from the septic wastewater, gave promising results regarding its use as AC. Chemical activation can be used to create very high surface areas, and the obtained results present their potential in surface area and porosity creation. KOH activation, which is a common activation method used in industry, produced high surface areas with similar porosities with both impregnation ratios (with 1:1 ratio 902 m^2^/g and 2:1 ratio 914 m^2^/g). Increasing the chemical amount in activation did not bring additional benefit in the surface area development but decreased the yield quite significantly (2.7% vs. 16% with the 1:1 ratio). The produced KOH activated carbons were very microporous, approximately 75% of total pore volume were micropores and 20% mesopores. High microporosity, combined with a sufficient amount of mesoporosity, are desired features in many purification applications, but also in higher value electrochemical applications. Generally, mesoporosity is required as a pathway to the micropores when aqueous solutions, e.g., electrolytes in electrochemical applications, are used. Activation using H_3_PO_4_ did not produce a high surface area. The relative meso– and especially macroporosity was very high (~57% of total porosity), which may indicate an excessively too high amount of reagent used in the impregnation.

#### 3.2.4. Techno-Economic Assessment

The mass and electric energy balances for both evaluated concepts were calculated based on the reduction and chemical consumption figures in each separation step obtained from the pilot scale experiments and data obtained from equipment suppliers. Table 4 presents the achieved total reductions (%) with the purifications system and the concentrations (mg/L) of TSS, COD, N total, and P total in the purified water fractions for all four evaluated cases.

##### Products and Discharge to WWTP

The concentrations of valuable nutrients in the raw fish processing wastewater are low. Even if their recovery, as separate streams, was technically possible, the main challenges in utilizing these by-product streams would be their low amount and the extra costs deriving from their possible concentration or transportation. The concentration of NH_4_-N in the raw septic tank wastewater is rather high. In this case, it would be justified to recover the nitrogen nutrient as a separate stream. Possible technologies, proved in the laboratory and pilot scale tests, would be adsorption or MC. However, both of these technologies are challenging from the feasibility and investment point of view. The purification system for the septic wastewater should be as manageable and automatic as possible, and should not require much labor. Both these technologies require handling of chemicals and adsorbents, which increases the complexity and costs. Moreover, the investment in the equipment is a heavy burden for a treatment unit with such a small annual capacity (560 m^3^/a). In all four evaluated cases, the most potential product stream together with purified water is the solid fraction from the belt filtration. It can be used for the production of hydrochar via HTC (SepticWW) or biogas via methanation (FishWW and SepticWW). Even the low dry matter (20% in cases SepticWW#1 and SepticWW #2) is not a hindrance for the HTC, where wet feedstocks may be utilized. In both concepts, the solid fraction may also be composted. In the TEA, it was assumed that there will be neither revenues nor waste handling fees deriving from the solid fraction.

In the FishWW#1 case, the target was to decrease the wastewater treatment fee by cutting down the COD level of the original wastewater before discharging it to a municipal WWTP. After the treatment, the total amount of the wastewater remained the same, but an 87% COD reduction was achieved. In all three other evaluated cases, the target was to purify the wastewater to fulfill the quality requirements of “pure water”. In case FishWW#2, about 60% of the wastewater was purified to “pure water”, whereas the remaining 40% still needs final treatment in a municipal WWTP. In the SepticWW#1 and SepticWW#2 cases, 45% to 50% of the original wastewater was treated to be discharged back to sea as such, whereas the remaining 50% to 55% still requires final treatment in a municipal WWTP. The challenge is the low volume reduction ratio (VRR) achieved in the RO unit(s). The lower the VRR the higher the volume of the RO concentrate directed to a municipal WWTP is, which results in additional costs. In the FishWW#2 case, a VRR of 2.5 was achieved. In the SepticWW#1 and SepticWW#2 cases, for the first pass of the RO, a VRR of 2 was achieved, and for the second pass of RO, the VRR was 10. In both concepts, it would be important to maximize the VRR, thus to minimize the amount of RO concentrate. The wastewater treatment fees deriving from the RO concentrate were excluded in the TEA.

##### Economics

Based on the process and cost analysis assumptions, the treatment costs for one m^3^ of wastewater to be treated in the purification system (EUR/m^3^ WW in) were calculated. The variable costs consist of electricity and chemical costs, cartridge filter and RO membrane replacement costs, and RO membrane cleaning costs. The total variable costs were 0.024 EUR/m^3^ WW in for FishWW#1, 0.77 EUR/m^3^ WW in for FishWW#1, 2.78 EUR/m^3^ WW in for SepticWW#1, and 2.04 EUR/m^3^ WW in for SepticWW#2. The most crucial variable cost contributor in the FishWW#1 case was the chemical costs (73% of total). In the three other evaluated cases, they were the specific electricity consumption in RO and the RO membrane replacement and cleaning. Their shares from the total variable costs were 8% to 30% and 36% to 46%, respectively.

The fixed costs, including labor, labor overheads, maintenance, taxes, and insurance, were: 0.32 EUR/m^3^ WW in for FishWW#1, 0.43 EUR/m^3^ WW in for FishWW#1, 22.0 EUR/m^3^ WW in for SepticWW#1, and 15.3 EUR/m^3^ WW in for SepticWW#2.

The total production costs (EUR/m^3^ WW in) are the sum of the variable and fixed costs. The total production costs, excluding the capital charges, and the share of different cost factors as a percent from the total production costs for all evaluated FishWW and SepticWW cases, are presented in Figure 12 and Figure 13. The cost structure indicates that the total treatment costs are very sensitive to labor costs (direct and indirect) and other fixed costs (maintenance, taxes, and insurance), which are the major cost contributor in all cases. In both FishWW cases, the labor costs were zero. The share of labor costs of the total treatment costs for SepticWW cases were 30–42%. The amount of other fixed costs was estimated as percentages of the FCI. Their share of the total treatment costs was 93% for FishWW#1, 36% for FishWW#2, 47% for SepticWW#1, and 58% for SepticWW#2.

The total purchased equipment cost (PEC) estimates for the whole wastewater treatment system were EUR 77,000, EUR 105,000, EUR 105,000, and EUR 91,000 for FishWW#1, FishWW#2, SepticWW#1, and SepticWW#2, respectively. They were based on the equipment cost estimations for the main equipment (including the belt filter, cartridge filter, and RO unit, but excluding pumps) obtained confidentially from equipment suppliers. Applying the installation coefficient of 1.55, the fixed capital investment (FCI) for the evaluated cases was EUR 119,000, EUR 162,000, EUR 162,000, and EUR 141,000.

##### Sensitivity Analysis

According to Figure 12 and Figure 13, the total treatment costs are very sensitive to the estimate of the other fixed costs (base case 4% of the FCI). The sensitivity analysis for the case SepticWW#2 shown in Figure 14 indicates that if the other fixed costs were only 2% of the FCI, the total treatment costs would decrease by 29% from 17.3 to 12.3 EUR/m^3^ WW in. To make the SepticWW#2 concept economically more attractive, the processing capacity should be much higher. The duplication of the capacity to 1120 m^3^/a, by maintaining the volume flow (1 m^3^/h) to the treatment system and doubling the operations hours (1120 h/a), the treatment costs would decrease by 31% from 17.3 to 11.9 EUR/m^3^ WW in.

## 4. Discussion

Biological treatments have attracted most of the attention in wastewater purification, whereas, so far, little attention has been paid to physico-chemical technologies. The physico-chemical technologies give, however, great possibilities to recover nutrients and carbon when purifying wastewater. In this study, the most potential physico-chemical unit operations for treating two different wastewater fractions were scaled up, installed, and piloted in a mobile container. The selected unit operations, belt filtration with an integrated screw, cartridge filtration, and RO, could techno-economically produce exploitable circular economy products and be assembled in a mobile container for carrying out recovery where suitably formed.

The results of the pilot trial conducted in the mobile container for fish processing wastewater showed that turbidity, COD, N total, NH_4_-N, P total, and PO_4_-P reductions were after the belt filtration of the flocculated wastewater 99%, 88%, 50%, 50%, 20%, and 18%, respectively. After the whole purification procedure, the rejection of all studied parameters was over 95%. The major products derived from the fish processing wastewater were pure water and carbon product made from the solids recovered from the screw during the belt filtration. The solid fraction (TS 50–85%) was treated by HTC. The major product of the HTC reaction was oil-like biocrude liquor. The produced hydrochar was not tested for activated carbon production due to the low hydrochar yield. P and N products were not obtained, since their original concentrations in the fish processing wastewater were too low for reasonable recovery. Pure water obtained from RO contained no solids, 1.1 mg/L N total, 0.1 mg/L P total, and 15 mg/L COD being good enough either for reuse or discharge.

Turbidity, COD, N total, NH_4_-N, and P total reductions were 51%, 50%, 8%, 10%, and 35%, respectively, after the belt filtration pilot trials of flocculated septic wastewater. The TSS of the suspended solids fraction recovered from the screw during belt filtration was 20%. The solids fraction was treated by HTC to produce hydrochar. The activation tests of the hydrochar gave promising results regarding its use as activated carbon. Chemical activation produced a large surface area and high porosity to the hydrochar. The whole purification procedure in a mobile container produced a rejection of over 95% to all measured parameters. Pure water obtained by means of two-pass RO contained no solids, 19 mg/L COD, less than 0.6 mg/L P total, and 20 mg/L NH_4_-N and N total. This was good enough quality for reuse purposes or discharge. The N product obtained from the RO concentrate was 44 g/L of (NH_4_)_2_SO_4_. The P product was not obtained due to its low concentration in the septic wastewater, 15 mg/L P total.

The techno-economic feasibility of both fish processing cases seemed promising. The total treatment cost for only cutting down the COD content of the fish wastewater by 87% was 0.34 EUR/m^3^ WW in and for purifying the fish wastewater for reuse was 1.20 EUR/m^3^ WW in. The main challenge is the variation of the fish wastewater’s quality. In the case of the septic wastewater, the total treatment cost for producing pure water varied between 17.3 and 24.8 EUR/m^3^ WW in. The main obstacle for the feasibility is the very small capacity to be purified: annually only 560 m^3^. Due to the small capacity, the labor and other fixed costs corresponded to almost 90% of the total treatment costs. The capital charges were not included in the treatment costs of either of the concepts. The total capital investment was, for the fish wastewater treatment system, EUR 119,000 to EUR 162,000 and for the septic wastewater treatment system, EUR 141,000 to EUR 162,000. The total treatment cost for producing phosphorous and nitrogen fertilizers was not calculated here, but it should be noted that the solids fraction obtained from the belt filtration is also a valuable stream in both evaluated concepts. It can either be used for the production of hydrochar via HTC (septic wastewater) or biogas via methanation (fish and septic wastewater). Even the low dry matter of the solids fraction (20%) is not a hindrance to the HTC, where wet feedstocks may be utilized.

Some suggestions for future research arose during the study. Higher water recoveries than achieved necessitate improvements especially in chemical treatment, MF and RO. When optimizing the coagulation and flocculation, pH adjustment of wastewater should be considered, since those are known to perform better at lower pH than applied [25]. Then, RO concentrate can be used as backwash to prevent clogging of MF filters [26]. In addition, NF before RO may prevent fouling of RO membranes by rejecting multivalent ions [27]. When NF-treated water is led to RO, it could operate at a higher recovery rate, producing more water and, in principle, reduce the OPEX of the operation. However, NF suffers fouling as well as RO. Thus, the overall water recovery when using the combination of NF and RO does not necessarily produce higher recovery than RO alone. When a second RO is needed, such as in the case of SepticWW#2, the concentrate from the second RO unit could be recycled to the feed of the first unit. With this, an increase in overall water recovery and, hence, reduced OPEX may be reached.

## Figures and Tables

**Figure 1 membranes-11-00975-f001:**
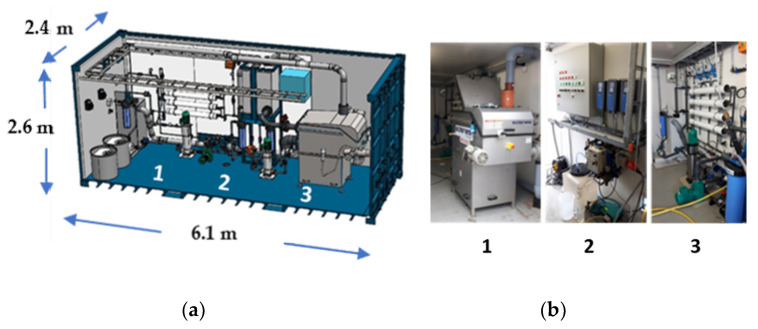
(**a**) A schematic of a 20-foot resource container constructed in the study: (1) Salsnes SF1000 belt filter with an integrated screw and a polymer feed system, (2) cartridge filters, and (3) a RO membrane unit; and (**b**) photos of the equipment: (1) Salsnes SF1000 belt filter with an integrated screw and a polymer feed system, (2) control center, and (3) a RO membrane unit installed to the container.

**Figure 2 membranes-11-00975-f002:**
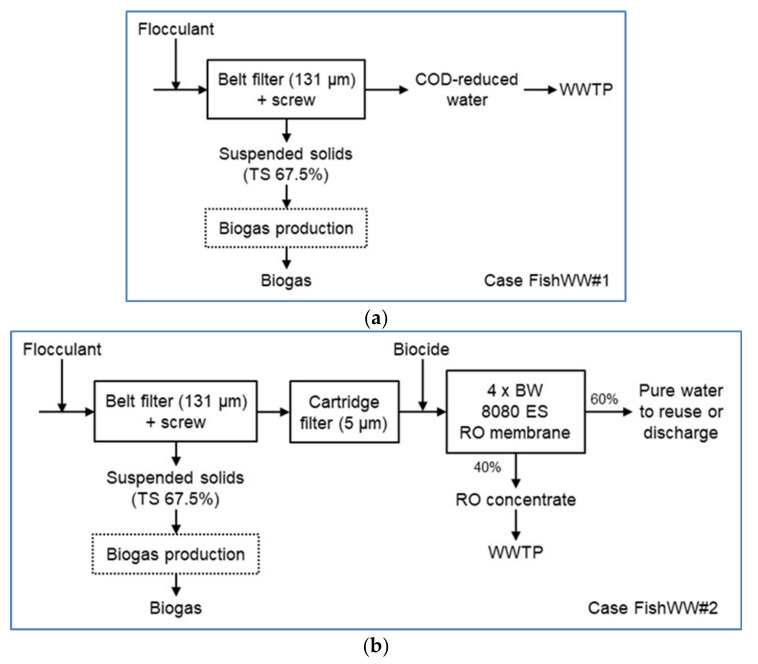
(**a**) Block diagram for cases FishWW#1 and (**b**) FishWW#2.

**Figure 3 membranes-11-00975-f003:**
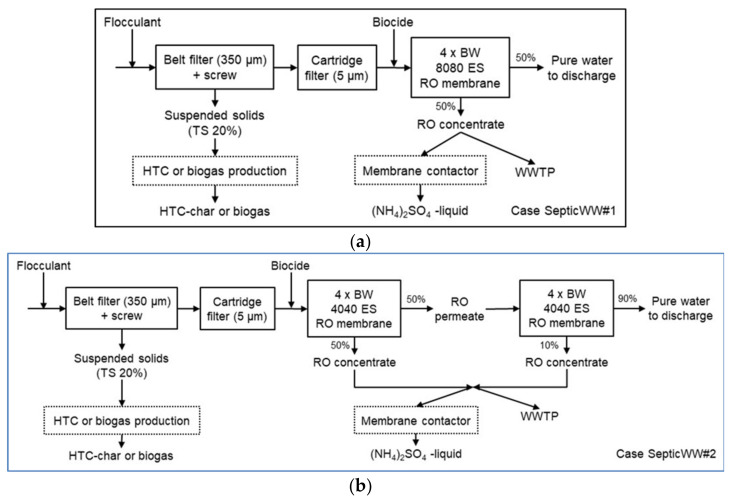
(**a**) Block diagram for cases SepticWW#1 and (**b**) SepticWW#2.

**Figure 4 membranes-11-00975-f004:**
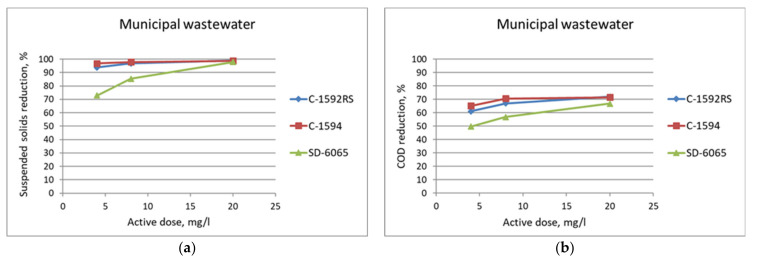
Reduction in suspended solids (**a**) and COD (**b**) using emulsion flocculants for the municipal wastewater at laboratory scale tests.

**Figure 5 membranes-11-00975-f005:**
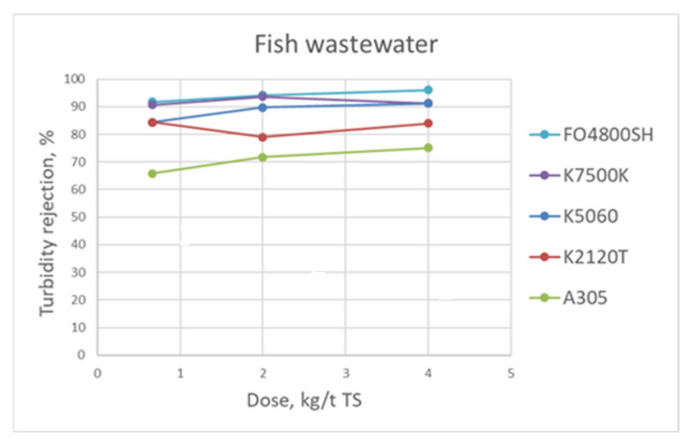
Turbidity rejection as a function of polymer active dose for the fish wastewater at laboratory scale tests.

**Figure 6 membranes-11-00975-f006:**
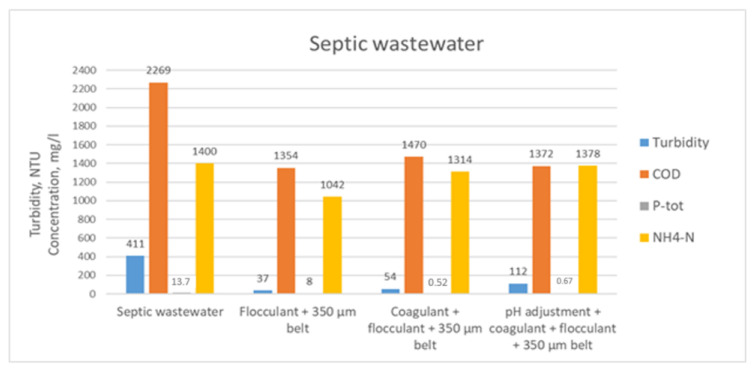
Turbidity rejection and reduction in COD, P-tot, and NH_4_-N in chemical treatments and belt filtration of septic wastewater at laboratory scale tests.

**Figure 7 membranes-11-00975-f007:**
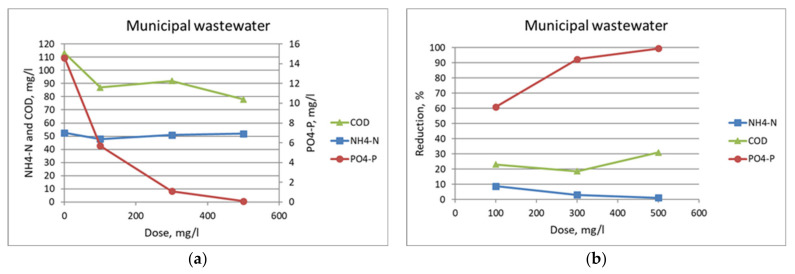
Effect of Ca(OH)_2_ dose on the concentrations (**a**) and reductions (**b**) of NH_4_-N, PO_4_-P, and COD for the municipal wastewater sample at laboratory scale tests.

**Figure 8 membranes-11-00975-f008:**
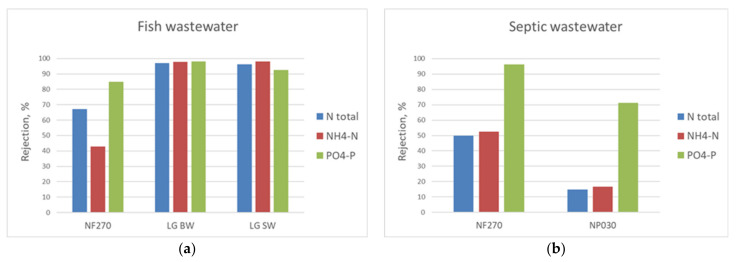
Nutrient rejection of selected NF (NF270, NP030) and RO (LG BW, LG SW) membranes for fish processing wastewater (**a**) and septic wastewater (**b**).

**Figure 9 membranes-11-00975-f009:**
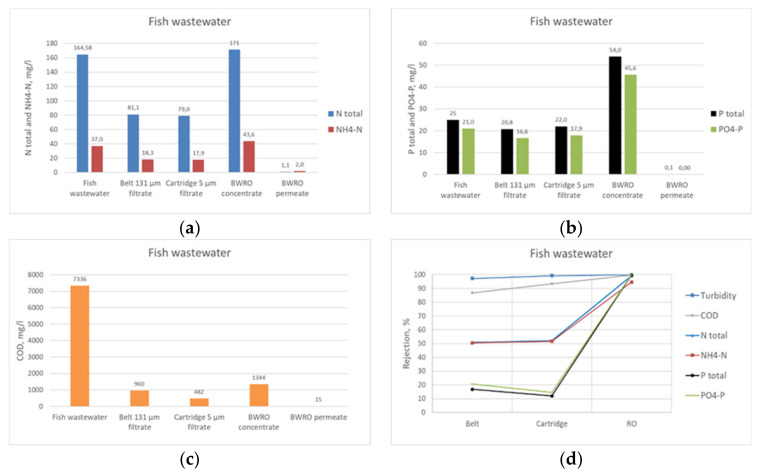
Results of piloting the fish processing wastewater: (**a**) N total and NH_4_-N contents, (**b**) P total and PO_4_-P contents, (**c**) COD contents, and (**d**) rejections of turbidity, COD, N total, NH_4_-N, P total, and PO_4_-P contents.

**Figure 10 membranes-11-00975-f010:**
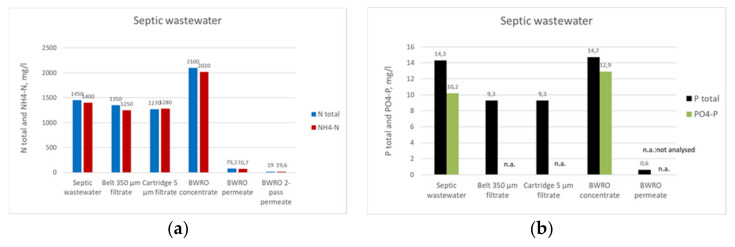
Results of piloting the septic wastewater; (**a**) N total and NH_4_-N contents, (**b**) P total and PO_4_-P contents, (**c**) COD contents, and (**d**) rejections of turbidity, COD, P total, N total, and NH_4_-N contents.

**Figure 11 membranes-11-00975-f011:**
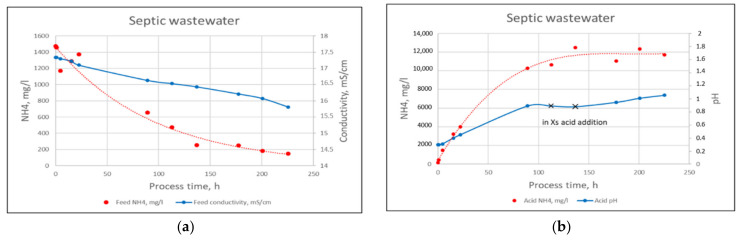
Conductivity and ammonia content of feed solution (**a**) and pH and ammonia content of acid solution (**b**) during a 225-h membrane contactor process with septic wastewater.

**Figure 12 membranes-11-00975-f012:**
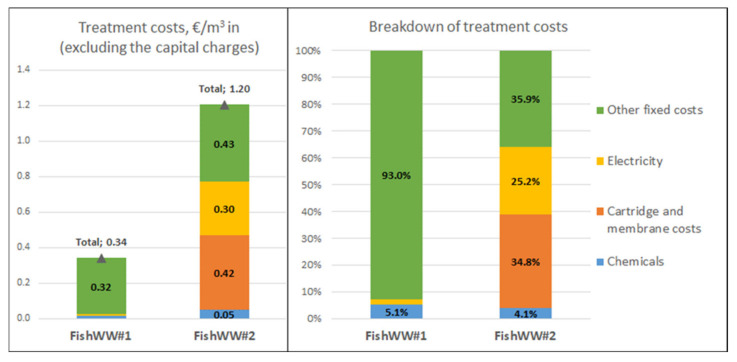
The total treatment cost estimates (EUR/m^3^ in) excluding the capital charges and the breakdown of the costs for the FishWW concepts.

**Figure 13 membranes-11-00975-f013:**
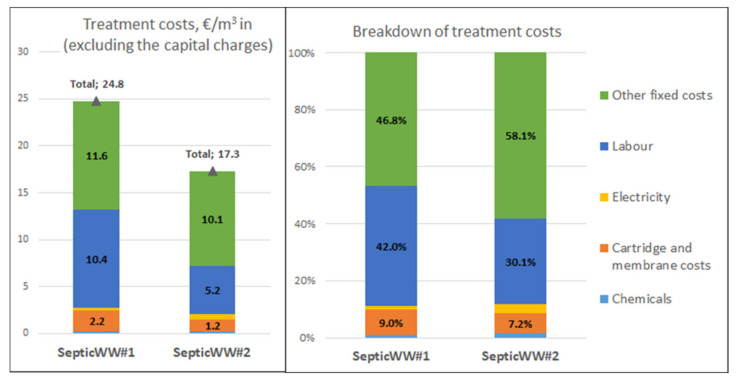
The total treatment cost estimates (EUR/m^3^ in) excluding the capital charges and the breakdown of the costs for the SepticWW concepts.

**Figure 14 membranes-11-00975-f014:**
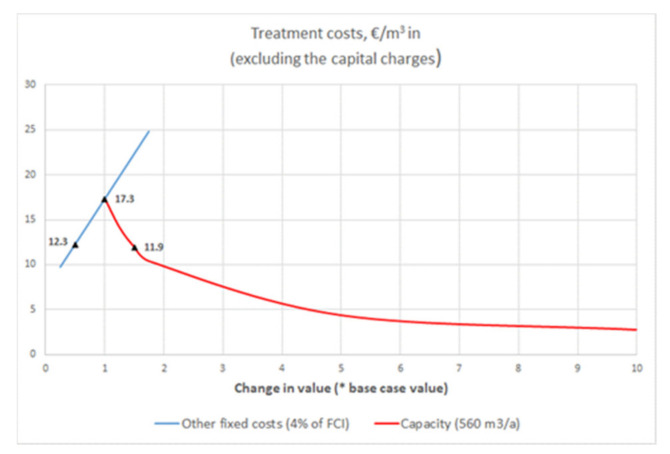
Sensitivity of treatment costs of case SepticWW#2 on selected process variables. The base case value of each parameter is shown in legend in parenthesis.

**Table 1 membranes-11-00975-t001:** Prices used for evaluating the variable costs.

	Price	Unit
Flocculant	2	EUR/kg
Biocide	2.5	EUR/kg
Electricity	70	EUR/MWh
Cartridge filter media unit price	10	EUR
RO membrane	25	EUR/m^2^
RO membrane cleaning cost	50	EUR/m^2^/a

**Table 2 membranes-11-00975-t002:** Factors used in the detailed factorial method for estimating the fixed capital investment (FCI).

Total Purchased Equipment Cost	∑PEC
f_1_ Equipment erection	0.5
f_2_ Piping	0.6
f_3_ Instrumentation and control	0.3
f_4_ Electrical	0.2
f_5_ Civil	0.3
f_6_ Structures and buildings	0.2 → 0
f_7_ Lagging and painting	0.1
f_8_ Material factor (stainless steel)	1.3
ISBL Cost C = ∑PEC * (1 + f_2_) f_8_ + (f_1_ + f_3_ + f_4_ + f_5_ + f_6_ + f_7_)	
ISBL Cost C = ∑PEC x	2.68
f_9_ Offsites	0.4 → 0
f_10_ Design and engineering	0.25
f_11_ Contingency	0.1
FCI = C * (1 + f_9_)(1 + f_10_ + f_11_)	
FCI = C x	1.35
**FCI = ∑PEC x**	**3.61**

**Table 3 membranes-11-00975-t003:** Characterization of the wastewater samples.

Wastewater	pH	Conductivity, mS/cm	TSS 0.45 µm, mg/L	COD, mg/L	N Total, mg/L	NH_4_-N, mg/L	P Total, mg/L	PO_4_-P, mg/L
Municipal wastewater	7.3	1.2	200	440	88	76	12.8	8.6
Fish processing wastewater	5.7	2.0	1940	7336	165	37.0	25.0	21.0
Septic wastewater	8.1	17.9	820	3689	1450	1400	15.5	10.9

**Table 4 membranes-11-00975-t004:** The achieved total reductions (%) and the contents (mg/L) of TSS, COD, N total, and P total in the purified water fractions for all four evaluated cases.

**Total Reduction, %**	**FishWW#1**	**FishWW#2**	**SepticWW#1**	**SepticWW#2**
TSS	86.4	100	100	100
COD	87.0	99.9	99.0	99.8
N total	50.9	99.6	97.3	99.4
P total	16.9	99.8	98.1	99.9
**Content, mg/L**	**FishWW#1**	**FishWW#2**	**SepticWW#1**	**SepticWW#2**
TSS	265	0	0	0
COD	960	15	77	19
N total	81	1.1	79	20
P total	21	0.1	0.6	0.02

## Data Availability

Not applicable.

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
