# Peer review of "Wastewater Purification with Nutrient and Carbon Recovery in a Mobile Resource Container"

_membranes, 2021, doi:10.3390/membranes11120975_

Round 1
Reviewer 1 Report
The paper presents a complete wastewater treatment unit based on only Physico-chemical technologies in a small platform (a mobile container). The concept and selected technologies are very interesting and were proved their performances in both lab and pilot-scale experiments in this study. The authors also report an economic analysis that will help readers better see the benefit of resource recovery. Therefore, I would like to suggest publishing this article and having only some minor comments to improve the quality of the work.
Line 28 “water is valuable product of wastewater” I understand the authors mean “Recycled water” instead of just water?
Line 63 The authors should clarify why the phosphorous is a part of awkwardly placed sewage sludge?
Line 70 The mobile container was first mentioned here, but it should be better introduced. What is its overall concept ?, purposes? Size? The other benefits?
2.1-2.6 The capacity of lab-scale experiments should be mentioned. What are the flow rates or volume of input wastewater?
2.7 and figure 1. It would be very interesting to see the actual photo of the unit here, and a reference scale should be added to the figure.
Line 420 what is the original source of the septic wastewater?
Line 426 Are there any price differences in these flocculation chemicals? Some inputs on the economic point of view would be great here.
Line 496. The pH > 11 or 12 mentioned here and in other parts are very high for the conversions of NH4+. Is there any reason to use such a high pH which will contribute greatly to the process cost? The authors also should indicate what the membrane used in this experiment is. Is it a commercial or an in-house one?
Line 513 Is there any reason why the dose of FO4800SH at 6 ppm is selected here? Is it the optimal condition for figure 5?
Author Response
Dear reviewer,
Thanks for your comments. Attached is our answers/comments.
Best regards,
Antti Grönroos

Reviewer 2 Report
Why do the author use polynomial regression to fit the curve in Figure 11?
Author Response
Dear reviewer,
Thanks for your comments. Attached you have our answers/comments.
Best regards,
Antti Grönroos

Reviewer 3 Report
Authors have conducted extensive research on nutrient and carbon recovery and wastewater purification, including the application of mobile resource container. However, despite dealing with various topics, the experimental conditions are not consistent. Above all, the purpose of this study is unclear and there is no conclusion. Overall, it is difficult to consider it as a research paper. I highly recommend dividing this paper into several papers on a unified topic and increasing the discussion.
Author Response

(The authors gave the same response as above.)

Reviewer 4 Report
Title: Wastewater purification with nutrient and carbon recovery in a mobile resource container
Dear authors,
The topic of your paper is interesting. Overall, the quality of this paper is adequate, however there are some aspects which should be improved.
Abstract, title and references:
The aim is clear.
The references are relevant, recent, and appropriate.
B.- Introduction:
The research question is clearly outlined.
C.- M&M:
The variables are defined and measured appropriately. Methods are valid and reliable.
D.- Results & Discussion:
Data in appropriate way. Figures and tables are ok.
Conclusions are supported by results or/and references. However, there is some aspects which should be improved: suggestions for future research.
It would be interesting some additional managerial implications in line with the findings of the study. Practical implications? Something to inspire future research or implications for practice.
Your research is consistent.
Author Response

(The authors gave the same response as above.)

Reviewer 5 Report
In this article, the authors demonstrate a wastewater treatment concept with physio-chemical unit operations, which separates water, nutrient and carbon products with good reuse possibilities. The research is well thought out, starting from a laboratory scale and scaling them up towards containerized operation, and the techno economic analysis at the end makes the research attractive. In general, the paper is scientifically strong and would be a good contribution to the scientific community; hence, I recommend its publications after some minor revisions.
Some suggestions follow:
- The authors mention that NF membranes were not considered since they did not efficiently reject the nutrients from the treated wastewater. However, they could be an effective source to prevent fouling on RO membranes and extracting bivalents and higher metals if they are used after the belt filter. NF membranes are known to remove pathogens1. If NF treated water is further used in the RO, it could operate at a higher recovery producing more water and in principle reduce the OPEX of the operation.
- The authors should use the RO concentrate for backwashing of cartridge filters, it has been demonstrated in the RO industry2.
- For the case SepticWW#2, the authors should recycle the concentrate from the 2nd RO unit to the feed of the 1st unit as it will reduce the concentration of feed and increase overall system recovery and hence reduce OPEX.
- Coagulation and flocculation are known to perform better at lower pH3, the authors should consider that as an option for pretreatment of the wastewater.
References:
1 Liikanen, R., Yli-Kuivila, J. & Laukkanen, R. Efficiency of various chemical cleanings for nanofiltration membrane fouled by conventionally-treated surface water. Journal of Membrane Science 195, 265-276 (2002).
2 Qin, J.-J., Oo, M. H., Kekre, K. A. & Liberman, B. Development of novel backwash cleaning technique for reverse osmosis in reclamation of secondary effluent. Journal of membrane science 346, 8-14 (2010).
3 Altmann, T. & Das, R. Process improvement of sea water reverse osmosis (SWRO) and subsequent decarbonization. Desalination 499, 114791, doi:https://doi.org/10.1016/j.desal.2020.114791 (2021).
Author Response

(The authors gave the same response as above.)

Round 2
Reviewer 3 Report
I don't think this paper has been well revised.
Nevertheless, I respect the opinions of other reviewers.
So I think the final decision is left to the editor.